# Reduction of SEM charging artefacts in native cryogenic biological samples

Abner Velazco[1], Thomas Glen[1], Sven Klumpe [2], Avery Pennington[1], Jianguo Zhang[1], Jake L. R. Smith[1,3], Calina Glynn [1], William Bowles[1,3,4], Maryna Kobylynska[5], Roland A. Fleck[5,6], James H. Naismith[7], Judy S. Kim [1,8], Michele C. Darrow [1], Michael Grange [1,3], Angus I. Kirkland[1,8] & Maud Dumoux [1] ✉

Scanning electron microscopy (SEM) of frozen-hydrated biological samples allows imaging of subcellular structures at the mesoscale in a representation of their native state. Combined with focused ion beam milling (FIB), serial FIB/SEM can be used to build a 3-dimensional model of cells and tissues. The correlation of specific regions of interest with cryo-electron microscopy (cryoEM) can additionally enable subsequent high-resolution analysis. However, the use of serial FIB/SEM imaging-based methods is often limited due to charging artefacts arising from insulating areas of cryogenically preserved samples. Here, we demonstrate the use of interleaved scanning to attenuate these artefacts, allowing the observation of biological features that otherwise would be masked or distorted. We apply our method to samples where inherent features were not visible using conventional scanning. These examples include membrane contact sites within mammalian cells, visualisation of the degradation compartment in the algae *E. gracilis* and observation of a network of membranes within different types of axons in an adult mouse cortex. The proposed alternative scanning method could also be applied to imaging other non-conductive specimens in SEM.

Volume electron microscopy (EM) comprises a group of electron imaging techniques that are used to visualise samples at different spatial resolutions. The spectrum of techniques generally incorporates imaging using either a transmission electron microscope (TEM) or a scanning electron microscope (SEM). For TEM imaging, the main restrictions are related to the sample thickness, which is limited by the electron mean free path[1,2]. SEM imaging gives lower resolution but can be used to image larger volumes when combined with sectioning techniques using either a diamond knife or a focused ion beam[3]. However, with secondary electron SEM imaging, samples can be subject to electrical charging during scanning, leading to the presence of artefacts in the images[4–6]. This charging depends on the electrical properties of the sample and can also depend on the sample preparation since the electrical conductivity can be modified by embedding or staining the sample[7,8]. Gas injection-based charge compensation and variable pressure SEMs have been also used to reduce charging[9–11] but have a negative impact on the spatial resolution and are incompatible with experiments under cryogenic conditions.

SEM acquisition parameters directly impact the dynamic charge-discharge behaviour in insulators[12], and can be optimised to reduce

[1]The Rosalind Franklin Institute, Harwell Science & Innovation Campus, Didcot, UK. [2]Research Group CryoEM Technology, Max Planck Institute of Biochemistry, Martinsried, Germany. [3]Nuffield Department of Medicine, University of Oxford, Oxford, UK. [4]Diamond Light Source, Harwell Science & Innovation Campus, Didcot, UK. [5]Randall Centre for Cell & Molecular Biophysics, King's College London, London, UK. [6]Centre for Ultrastructural Imaging, King's College London, London, UK. [7]University of Oxford, Mathematical, Physical and Life Science Division, Oxford, UK. [8]Department of Materials, University of Oxford, University of Oxford, Oxford, UK. ✉e-mail: maud.dumoux@rfi.ac.uk

charging effects for a particular sample[13]. For example, reducing the accelerating voltage (to find a crossover energy at which the deposited and emitted charge are balanced) has been shown to reduce the buildup of charge[14–16]. Alternatively, reducing the dwell time (irradiation time per pixel) and increasing the distance between scanning points can also mitigate charging artefacts[17–19] by reducing the local rate at which the charge is deposited in the sample. However, all these approaches have limitations. Reducing the voltage, beam current or the dwell time can reduce the signal to noise ratio (SNR) in the images and may require the use of dedicated electron optical and detection systems, whilst increasing the distance between scanning points by simply reducing the magnification lowers the resolution.

In many cases, images are not acquired with the full electron fluence per scanning position in one pass. Instead, the fluence is distributed across several line scans or in multiple frames before integration of the signal[17,19]. Integration of the signal line-by-line avoids potential issues with drift but deposits the electron fluence in a limited area (a line), which can lead to charge build up and cause directional charging artefacts[20]. Frame integration is also subject to drift due to the increased time required to scan the same position more than once and requires registration of successive frames.

For insulating samples, these steps are often not sufficient to reduce charging artefacts[20–22]. Hence efforts have been made to computationally correct for charging artefacts, by using predictive models to subtract these and to recover the underlying signal[20,23]. However, directly reducing charging artefacts during imaging is preferable. Importantly, biological samples are inhomogeneous, adding further complexity to optimising beam and imaging conditions. The most common charging artefacts observed in vitreous biological samples are dark streaks around high-contrast structures such as lipid droplets (LD) or myelinated axons[20,24], and inhomogeneous contrast manifest as extremely dark and bright pixels[21]. Importantly, existing schemes that reduce charging artefacts in one region of an inhomogeneous sample may reduce the SNR or sharpness in other regions that do not experience these effects.

In this work, we describe an approach to mitigate charging artefacts by using an interleaved, or leapfrog, scanning pattern[25,26]. This approach distributes the electron fluence differently, in time and space, compared to conventional raster scanning, and allows for a more efficient charge dissipation. By applying image registration correction (MotionCor2[27]) to correct for beam and mechanical induced motion, frames from sequential acquisitions are aligned and integrated to improve the SNR. Using different biological samples, we compare image acquisition strategies, raster scanning with frame integration, raster scanning with line integration[20,24,28–30] and interleaved scanning with frame integration. We show that interleaved scanning allows the imaging of cells and tissue in a near-native state with minimal charging artefacts. The use of interleaved scanning also enables the observation of biological features otherwise obscured by these artefacts, specifically the local environment of LD in a mammalian cell model, the complex layers of the axon in mouse brain and the organisation of the network of degradative compartment (DC) in *Euglena gracilis*.

## Results

### Experimental setup

SEM images are typically acquired using raster scan patterns, where a line is imaged in one direction, often the $x$ axis of a rectangular coordinate system, typically from left to right (Fig. 1A, C), followed by the acquisition of the next line with a shift along the $y$ axis. In this sequence, the time interval between each position in the $x$ axis (fast scanning direction) is equal to the dwell time. While in $y$ (slow scanning direction), the interval is equal to the product of the dwell time and number of pixels along the $x$ axis. In contrast, interleaved scanning skips adjacent pixels in the $x$ and $y$ directions. Specifically, the

interleaved scan pattern used in this work skips 2 pixels in the $x$ direction, returns to the starting $x$ coordinate while skipping 2 pixels in the $y$ direction and repeats until all positions in the array have been visited (Fig. 1B, D). This interleaved configuration allows a longer time interval between subsequent $x$ or $y$ positions than for an equivalent raster scan. A linear pattern has been chosen as previous work demonstrates good performance with regards to image distortion in scanning-TEM (STEM)[31] while the skipping of 2 pixels in $x$ and $y$ has demonstrated damage reduction in high resolution STEM imaging[26].

For all scanning strategies the spatiotemporal fluence distribution (raster or interleaved) is convolved with the averaging mechanism used to form an image. We hypothesise that in 2 dimensions the spatial charge distribution is bigger than the beam size (estimated to be a 2 nm FWHM in our experiments) and that the charge dissipates uniformly in all directions from this point, assuming the sample has isotropic electrical properties[32–34] (Fig. 1E, F). Under this assumption, line integration accumulates charge locally as the beam is constantly rastered along a line, whereby charge can dissipate more efficiently in the $y$ direction but not in the $x$ line direction as this is constantly scanned (Fig. 1E). For frame integration, the charge does not accumulate in a local area but across the entire frame as charge dissipating from one position will accumulate during the exposure of neighbouring positions visited consecutively. Hence, for a raster scan the charge dissipation is not homogeneous, whereas for interleaved scanning, the charge dissipation is more uniform along the $x$ and $y$ axes (Fig. 1F), which is consistent with previous reports that show charging and beam damage[35] reduction for room temperature SEM and STEM[18,26]. However, we note that the interleaved scan described in this work is only one of many options to optimally engineer data collection strategies for insulating SEM specimens.

As biological exemplars, we report data from a mammalian cell model (RPE-1), a single-cell algae (*E. gracilis*) and mouse cortex. The mammalian cells and algae were vitrified by plunge freezing and the tissue was vitrified by high-pressure freezing (HPF). No samples were chemically fixed, dehydrated or stained. For plunge freezing, contact with a thermally and electrically conductive support (gold or carbon) is important for charge dissipation during imaging[36,37]. In contrast, during HPF of tissue, a thick tissue biopsy was deposited in a gold coated copper carrier filled with cryoprotectant before vitrification. Thus, the sample has limited direct contact with the carrier, reducing the electrical conductivity and the efficiency of charge dissipation.

These diverse biological samples of inherently different molecular assemblies as well as varied sample preparation methods have been explored to test if engineered scans are generally beneficial since previous reports show that some components in the cell are more prone to charging artefacts[20,24,28,29]. In particular, lipid droplets, stacking of membrane (thylakoid membrane in chloroplast, myelin sheets in brain) and degradation compartments (endosomes and lysosome) are often local charging centres due to their molecular composition[20,24,29]. Using different scanning strategies, we confirm that charge redistribution and available dissipation times during scanning can mitigate SEM charging artefacts in all the systems studied.

### Scanning pattern and dwell time

In our previous work[20,38] we demonstrated that using an Ar plasma for sputtering followed by secondary electron imaging at low kV (1.2 kV) it was possible to image volumes of vitrified biological samples at -10 nm resolution. In the present study we define an accelerating voltage and set a constant electron fluence ($10^{-1}$ e$^-$/Å$^2$). This electron fluence was distributed using three combinations of dwell time and number of scan repetitions to vary the pixel flux in three regimes as: low (100 ns ×100 repetitions), intermediate (500 ns ×20 repetitions) and high (1000 ns ×10 repetitions). The repetitions were completed either at the line integration (LI) or frame integration (FI) level and the scan pattern was either raster or interleaved. The angle between the electron beam

and the sample influences the image quality. When imaging at an angle, often 52°, the total secondary electron yield is optimal for the instrument used but differences in the electrical properties of the sample are lost leading to a reduction in the image contrast[39]. Consequently, imaging perpendicular to the milled surface increases the contrast[20,39]. As both imaging geometries are commonly used, we imaged the samples with the SEM column at 52° (Fig. 2 and Supplementary Figs. 1–3) and 90° (Supplementary Figs. 4–6) relative to the milled surface of the sample. After the region of interest (ROI) was exposed to $10^{-1}\,e^-/\text{Å}^2$, the surface was polished with an Ar plasma with a 50 nm step to avoid accumulation of electron beam damage during multiple acquisitions.

Our data show that for all fluence distributions and imaging angles considered, images acquired using raster scanning with line integration are dominated by streaks in the fast scanning direction (Fig. 2A. Supplementary Figs. 1A–C, 2A–C, 3A–C, 4A–C, 5A–C and 6A–C). However, an increase in dwell time (1000 ns) improved only some patches in the image where biological features could be discerned without artefacts (Supplementary Figs. 1C, 3C and 5C), highlighting the difficulty in optimising SEM imaging conditions for inhomogeneous samples. Using frame integration with a raster scanning pattern, images obtained at a glancing angle are corrupted by charging artefacts in the form of dark areas covering the majority of the scanned ROI. For longer dwell times (500 ns and 1000 ns), localized bright charging artefacts were also observed (arrows in Supplementary Figs. 1E, F, 2E, F, 3E, F, 4E, F, and 5E, F). When using an interleaved scanning pattern in combination with frame integration we observed that the charge accumulation is dependent on the dwell time irrespective of the imaging angle, with a 100 ns dwell time image showing minimal charging artefacts (Fig. 1C and Supplementary Figs. 1G–I, 2G–I, 3G–I, 4G–I, 5G–I and 6G–I). In comparison to raster scanning, HPF vitrified mouse cortex imaged with an interleaved pattern shows multiple membrane compartments isotropically resolved within regions that were not previously observed. This included the whole myelin sheath (green arrow Fig. 2D) and internal membrane compartments (the inner tongue) (pink arrow Fig. 2D). Resolving the internal membrane near the lipid rich myelin, which is particularly prone to charging, demonstrates the improvement using interleaved scanning for reducing charge artefacts, enabling the visualisation of additional details within frozen hydrated samples. Other imaging strategies (Fig. 2A, B) result in

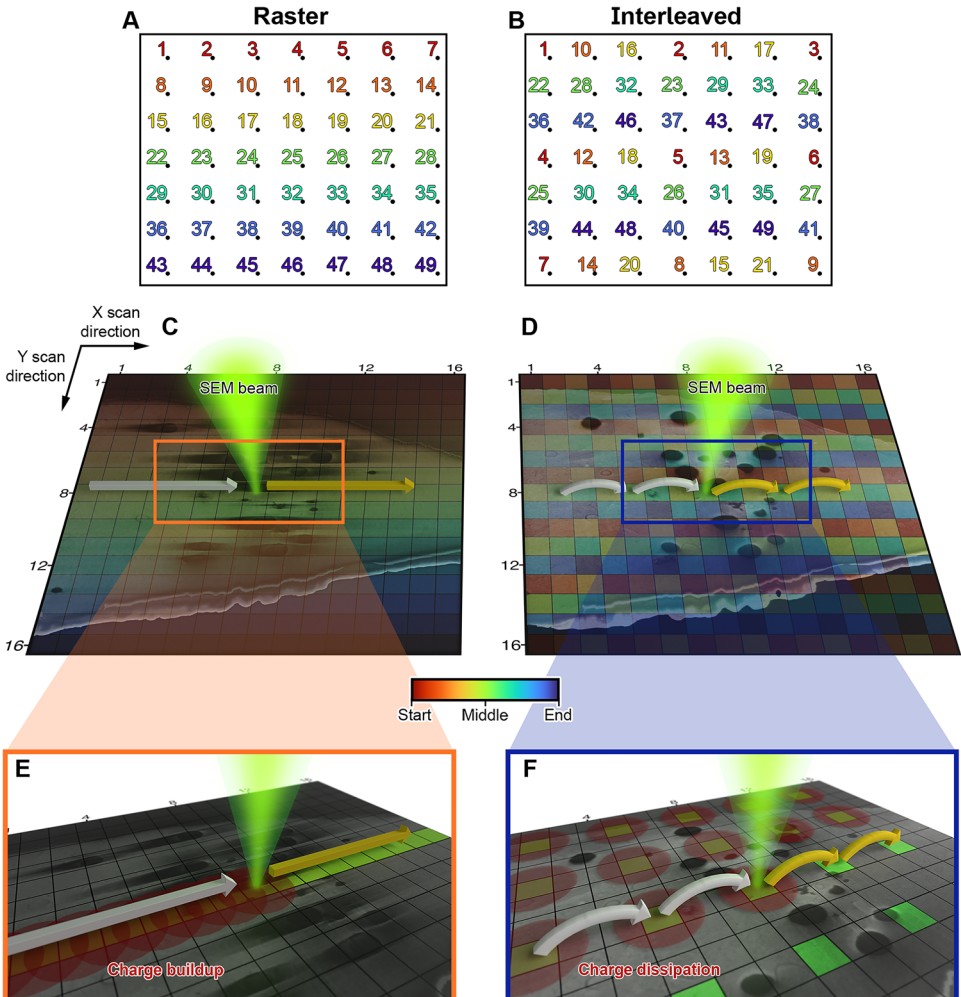

**Fig. 1 | Schematic of different scanning patterns. A, B** Field of view (black rectangle) imaged using a 7 × 7 array where the rainbow colour corresponds to the temporal distribution of the fluence (red at the start, blue at the end of the time sequence). **A** Raster scanning. **B** Interleaved scanning which skips 2 positions in *x* and 2 positions in *y*. **C–F** Overlay of vitrified RPE-1 with a schematic representation of pixel positions (black squares) in a 16 × 16 array. The green cone represents the electron beam. The straight arrows represent raster scanning while the curved arrows indicate pixel skipping during interleaved scanning. The colour of the arrow represents the pixel acquired (white) or yet to be acquired (yellow). **E, F** Spatial distribution of the electron fluence and isotropic charge dissipation (red). Yellow arrows indicate pixels have been imaged; green arrows indicate pixels yet to be imaged. Red circles illustrate charge dissipation radiating outwards at each position.

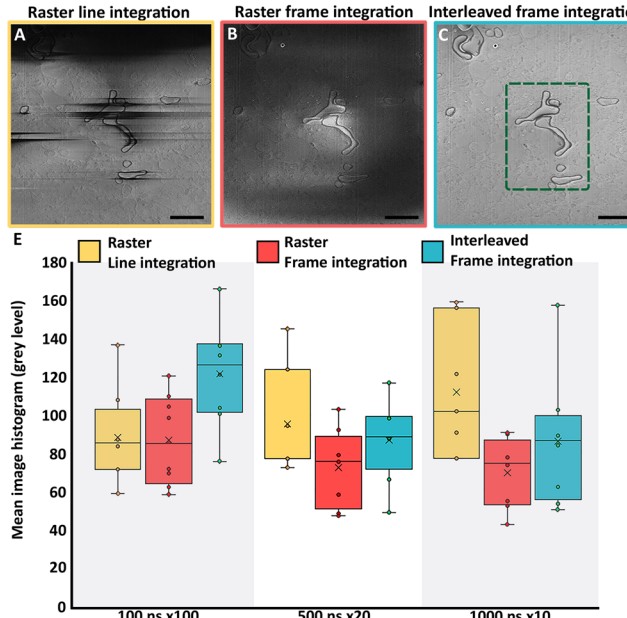

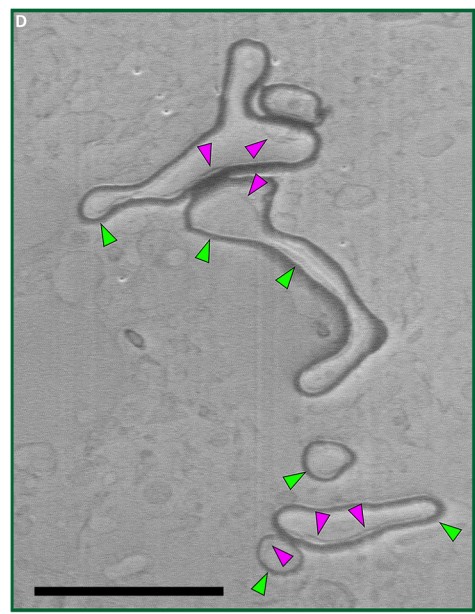

**Fig. 2 | Effect of electron fluence distribution on charging artefacts (SEM imaging at 52° angle to the FIB milled sample plane). A–C** Representative images of brain tissue for different pixel fluence distribution strategies using raster or interleaved scan patterns. **D** Enlarged area highlighted in (**C**) by a green dashed rectangle that shows detail visible within the membranous layers of myelination in the brain. Green arrows indicate the myelin sheath while pink arrows indicate the oligodendrocyte inner tongue. **C, D** Interleaved frame integration at 100 ns dwell time ×100 repetitions shows the greatest improvement in reducing charging artefacts. Scale bar: 2 μm. **E** Mean of image intensity histograms of different images (population n > 5, where n represents the number of acquisitions at the same scan condition over different samples. The exact size of the population can be found as each circle is a n.) from vitrified RPE-1 cells, *E. gracilis* and mouse brain images for raster line integration (yellow), raster frame integration (orange) and interleaved frame integration (blue). Population median (middle line), population mean (cross), median of the bottom half (bottom box), median of the top half (top box). Vertical lines extend to minimum and maximum intensity values. Full statistical analysis is given in Supplementary Table 1. Images acquired using interleaved scanning with frame integration using a short dwell time and high integration (100 ns dwell time ×100 repetitions) form a dataset for which the mean intensity is 122, closest to 127, which is the mid-range histogram intensity value of the 8-bit image data. Deviation in intensity from this value indicates charging as described in the text.

significant artefacts that obscure these details. We further note that previous work investigating the myelin sheath required resin embedding and staining[40–42].

The quality of an image can be characterised from the pixel intensity histogram (Supplementary Fig. 7). Typically, an image with minimal artefacts and high contrast features of interest would have a mid-range grey level Gaussian distribution of intensity, (127 for 8-bit images Supplementary Fig. 7G). However, any accumulation of charge visible as extreme dark or extreme bright pixels causes an asymmetry in the distribution (Supplementary Fig. 7A–F, H, I). We have compiled our data and extracted for each image the mean of the histogram. Some variation in the mean from sample to sample is expected for a dataset of high quality images (images without artefacts), the population concentrates around the mid-range scale. Deviation from this implies that the dataset is affected by large numbers of extreme pixel intensities, resulting from poor image quality (image with artefacts). For both imaging angles considered, the use of an interleaved pattern with frame integration using the low pixel flux strategy performed best with a pixel intensity centred around a 122 grey level when imaging 52° to the milled surface (Fig. 2E) and 97 when imaging perpendicular to milled surface (Supplementary Fig. 4). All the other imaging conditions give datasets with a lower mean (Fig. 2E and Supplementary Fig. 4J). We hence conclude that imaging with interleaved scanning using a short dwell time reduces charging artefacts, for both sample surface angles and for all biological sample types examined.

## Scanning strategy for automated image segmentation
Biological volume data acquired using SEM images is often post-processed using segmentation algorithms to highlight features relevant to a specific biological question (for example tissular/cellular organisation, volume analysis or length of contact)[43–45]. However,

curtaining[46,47] and charging artefacts are often a source of poor segmentation outcomes[20,48]. Consequently, various approaches have been used to computationally remove both types of artefacts[49,50]. Hardware implementations, such as specialised sample stages (rock milling stage) and the use of a plasma FIB argon source, have also been employed to reduce curtaining artefacts[20,30,51–53]. We have investigated whether the implementation of interleaved scanning benefits automated segmentation. We used the Segment Anything Model (SAM)[54] on the shortest dwell time (100 ns) datasets for raster line/frame integration and interleaved frame integration with minimum user input and the same set of parameters (see section "Methods"). The number of picked objects were counted as well as the complexity of their shape. The overall aim in this analysis is to observe the relationship between the reduction of charging artefacts and unsupervised segmentation rather than to obtain the optimum segmentation for a defined component. This analysis was applied to mouse brain tissue, RPE-1 cells and *E. gracilis* imaged at 52° or 90° with respect to the milled sample plane. Supplementary Fig. 8 shows a representative result obtained from the segmentation. The absence of artefacts in data acquired using interleaved frame integration allowed more objects to be picked (Supplementary Fig. 8A–F). These objects were not deformed or attributed to charging artefacts (as is the case in Supplementary Fig. 8A, arrows).

Datasets of RPE-1, *E.gracilis* and mouse brain acquired at the same angle were merged to measure the number of objects picked using SAM and their complexity defined as a ratio between the squared perimeter and the area. As each dataset has a unique biological content, the value of the number of objects picked or complexity score was normalised to that calculated from the corresponding interleaved frame integration. Interleaved scanning images systematically allowed SAM to identify and locate more objects than the rastered images

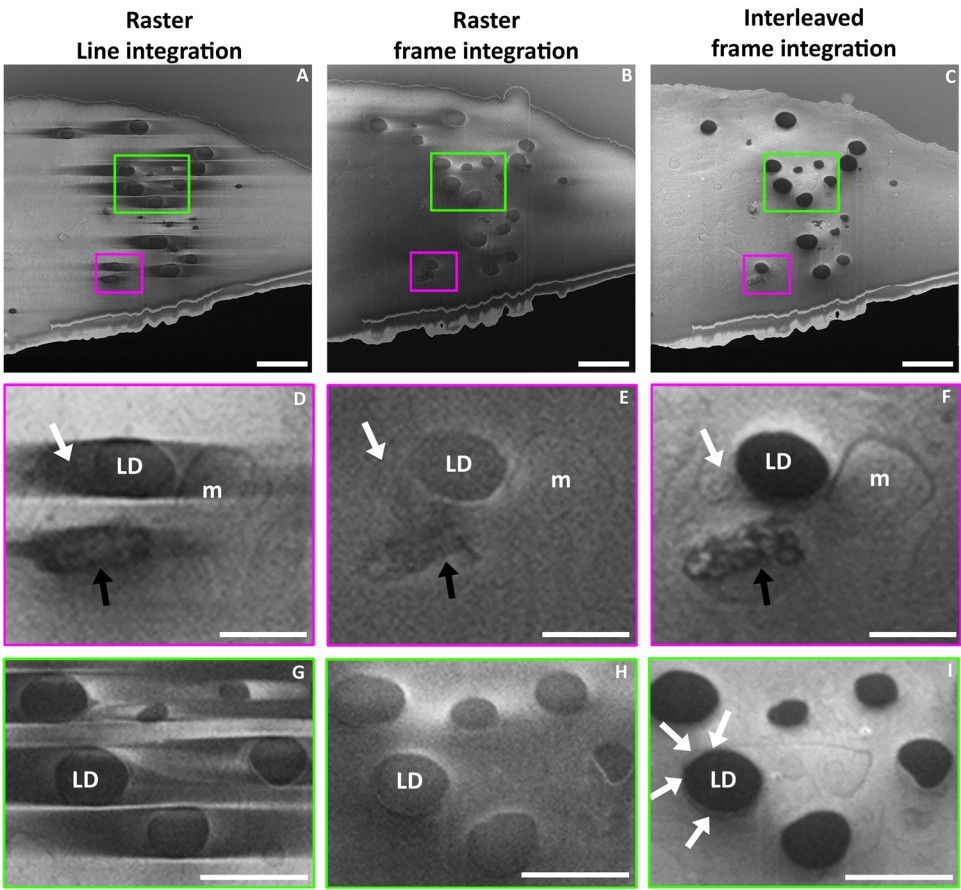

**Fig. 3 | Charging artefact reduction in RPE-1 cells allows the observation of lipid droplet (LD) surrounding environments.** Vitrified cells imaged at 52° with respect to the FIB milled sample plane using a 100 ns dwell time x100 repetitions. **A–C** Overview of a slice from RPE-1 imaged using raster line integration, raster frame integration and interleaved frame integration, respectively. Scale bar: 2 μm. **D–F** enlargement of the pink boxes in (**A–C**). White arrow: membrane in the vicinity of a LD. Black arrow: content of a degradative compartment. m: mitochondria. Scale bar 0.5 μm. **G–I** Enlargement of the green boxes in (**A–C**); brightness and contrast were modified to improve visualisation. White arrow: wrapping of endoplasmic reticulum (ER) around LD. Scale bar 1 μm.

(Supplementary Fig. 8G). Imaging using raster scanning frame integration produces images with artefacts that reduce the object picking by half on average. Only imaging using raster line integration at 52° allowed object picking with a value close to the reference (interleaved frame integration) with 80% of objects picked. In addition to the decrease in the number of objects picked by SAM when charging artefacts are present, the segmentation often misrepresents their actual shape[20,24,28,29,43,45]. Our analysis demonstrates that the objects picked by SAM from images acquired with raster imaging (frame and line integration) have a lower complexity score (20 to 40% less depending on the conditions), than those picked from images acquired with interleaved scanning (Supplementary Fig. 8H). Overall, we conclude that a combination of interleaved scanning with short dwell times and frame integration allows the imaging of vitrified biological samples with minimal charging artefacts, subsequently enabling improved segmentation of heterogeneous biological features.

## Dissecting the organelle landscape surrounding lipid droplets

When imaging mammalian cells, the presence of LD often leads to very significant charging artefacts and obscures the proximal detail of their local environment. Furthermore, using classical fixed, stained and embedding procedures for imaging at room temperature leads to artefacts, raising ambiguity about the identification of features and the relevance of the data[55,56]. Therefore, imaging vitrified samples is currently the most appropriate technique to study LD biology but as observed in Fig. 3 and Supplementary Figs. 2, 3, 5 and 6, LD act as

strong charging centres, making an analysis of their environment and interactions with other organelles difficult[28,38]. Using traditional raster methods (Fig. 3D, E), the neighbouring environment around the LD is not clear, the ER (white arrows) is not visible and the mitochondria are not well defined. Furthermore, the inside of the degradation compartment (black arrows) is obscured by charging artefacts. With interleaved scanning it is possible to observe mitochondria distinctly including cristae, possible ER membranes and a degradation compartment within the proximity of a LD (Fig. 3F). In addition, the image acquired using interleaved scanning allows the internal structure in the degradation compartment to be observed as an interconnected network of electron dense material (black arrows). These observations are not possible or are severely limited in images acquired using other scanning strategies.

LD are critical lipid sources for organelles that form MCS with the surface of the LDs[57,58]. The additional information in images formed using interleaved scanning allows the observation of endoplasmic reticulum (ER) wrapped around LD (Fig. 3G–I), suggesting a MCS, demonstrating that the alternative scanning strategy has a positive impact on studies of the interaction of LD with other organelles in native conditions.

## Analysis of the degradation compartment in *E. gracilis*

Unicellular organisms, such as *E. gracilis*, can control complex functions, such as motility, hunting, and digestion through specialised sub-cellular structures. Hence observation of the relevant membranes of key sub-cellular compartments is critical to understanding how algae

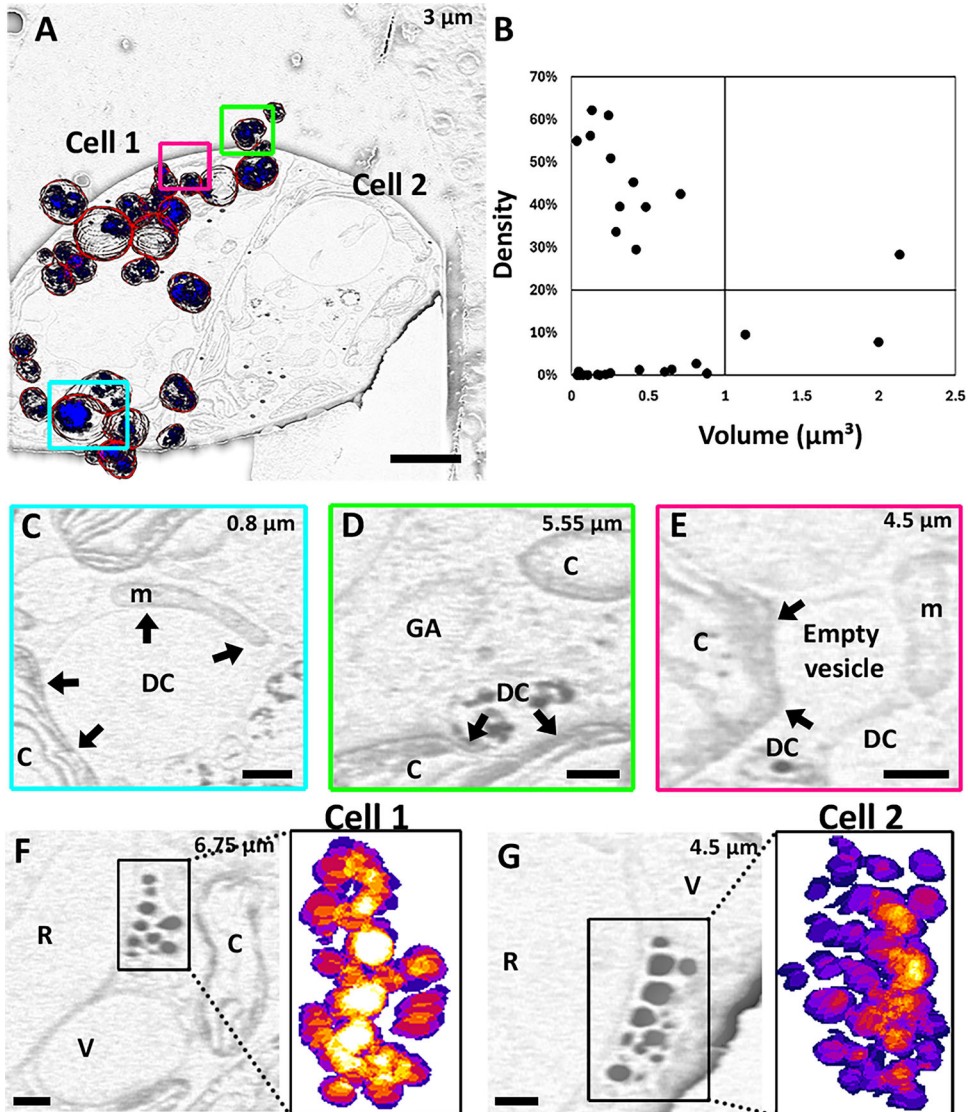

**Fig. 4 | SEM volume imaging of *E. gracilis*.** *E. gracilis* imaged at 52° with respect to the FIB milled sample plane using 100 ns dwell time ×100 repetitions. A 1600 µm³ volume in focus aligned and subject to manual segmentation of the region of interest. **A**, **C**–**G** Images after background removal (see "Methods") to assist segmentation. For **A**, **C**–**G**, the number in the upper corner is the z-location within the volume. **A** Membranes of the degradation compartment segmented (red line) and content (blue line). Scale bar: 2 µm. **B** Density (ratio of the content volume within the degradative compartment) as a function of the volume of the degradative compartment showing the presence of four unrelated populations (quadrant count ratio = 0.037). **C**–**E** Enlarged panels from the coloured boxes in (A), highlighting the membrane deformation of organelles in close proximity to the degradation compartment (DC) or empty vesicles and other organelles including mitochondria (m) and chloroplast (C) or an absence of contact with, for example, the Golgi apparatus (GA). The limits of these contacts are indicated by a black arrow. Scale bars: 500 nm (C), 250 nm (D–E). **F**, **G** Enlarged panels of the eyespot and 3D segmentation, proximal to the reservoir (R). Compartments in proximity (V: vesicle, C: chloroplast). Scale bar: 250 nm.

control these functions at a sub-cellular level. Supplementary Fig. 9 shows a sectional plane near the anterior end of the algae, close to the insertion point of the flagella. Imaging using interleaved scanning improved the observation of features in the vicinity of the reservoir and surrounding structures (Supplementary Fig. 9A–C), including paramylon and the membrane of the reservoir itself. We also observed that interleaved scanning is necessary to observe the thylakoid membrane sheets, the ER, neighbouring LD, vesicles contacting the reservoir as well as the membrane neighbouring the contractile vacuole (Supplementary Fig. 9D). Even in areas where charging artefacts are less common (Supplementary Fig. 9G-I, K, L), images recorded using interleaved scanning reveal additional details in the ultrastructure. For example, in Supplementary Fig. 9J, the membranes separating the contractile vacuole and the reservoir are clearly visible and in Supplementary Fig. 9K, the membrane delineating the degradation

compartment is visible. Similarly, in Supplementary Fig. 9L, the presence of LDs does not prevent the observation of the neighbouring chloroplast and associated thylakoid membrane.

It was further possible to reconstruct in three dimensions the network of degradation vesicles after volumetric acquisition (1600 µm³) of *E. gracilis* cells (Fig. 4 and Supplementary Movie 1). From observations of 55 vesicles only two are in proximity (<500 nm) to the reservoir and contractile vacuole, while the remainder form densely packed networks surrounded by paramylon and chloroplasts. Segmentation of the degradative compartment (DC), and its respective vesicular content, in one of the cells (Fig. 4A) shows that all the DC are different. The DC is equivalent to the digestive system in multicellular organisms, and by understanding their structure, it is possible to gain insight into their dynamic functionality. By plotting the volume of the vesicle as a function of the density of its content, (Fig. 4B) we can

distinguish the different vesicle populations (quadrant count ratio of 0.037). Most of these have a small volume (23/26 are under $1\,\mu m^3$) and within this population, half have 20% or more of the total volume filled with dense content. This suggests that these different vesicles are at different states in their development. We have also observed the environment of these different populations and noticed the presence of extended contact between the nucleus, mitochondria or chloroplasts and the DC, but are not yet able to differentiate between the two membranes of the organelles (Fig. 4C, D). These contacts could be mapped as it was possible to observe the deformation of the membrane of the organelles indicative of the presence of structured MCS between DCs and the chloroplast, nucleus or mitochondria. We did not observe specific extended contact with the Golgi apparatus, the reservoir or contractile vacuole. The DC forming these MCS are part of the low internal density population suggesting that the content of the DC may be used by other organelles. This hypothesis is supported by the observation of emptied vesicles in close proximity to DC (Fig. 4E) and we propose that these empty vesicles are emptied degradative compartments. Another specialised compartment that was characterised is the eyespot (Fig. 4F, G). Eyespots are formed by carotenoid globules close to the reservoir, whose role is to detect and direct the cell depending on the light source, therefore allowing the appropriate amount of light for the algae to develop[59]. To date, studies of the eyespot have been limited to correlative approaches using fluorescence and resin-embedded electron microscopy[60]. Using volume imaging, we were able to image these structures in their integrity and model in 3-dimensions the organisation of their granules using a single approach (Fig. 4F, G). This aspect of the work reported demonstrates that by engineering the scanning strategy it is possible to image single cellular organisms and analyse the intracellular organisation of key components of their life cycle.

## Classifying the neuronal network of an adult mouse brain

Finally, one of the major challenges in volume cryo-SEM imaging is the imaging of tissues, among which brain, rich in myelin, remains a challenge. Indeed, serial SEM analysis of brain is often limited to young individuals where myelinisation is not fully established[20,28,40,44]. Moreover, the ability to study the thickness of the myelin in the case of normal or pathological development has been limited by artefacts due to the fixation, dehydration and staining required for room temperature analysis[61] or the requirement for specialised sample preparation[62]. However, using modified scanning we have observed the myelinated structure in an adult brain cortex in a representative near-native state (Supplementary Fig. 10). We observed axons with thick or thin myelination (Ax or ax respectively) in all areas within the field of view, including areas where the axons were bundled. The reduction of charging artefacts also allowed the observation of structure within the axon (inner tongue of the oligodendrocyte, mitochondria, vesicle) and in their local vicinity (mitochondria, vesicles, ER).

The acquisition of a volumetric dataset using interleaved scanning and a short dwell time provides insights into myelination organisation (Supplementary Movie 2). In the central nervous system (CNS), myelin is formed by oligodendrocyte cells wrapping around the axon. It has been shown that the wrapping of the myelin sheath is a combination of two coordinated events; the wrapping of the leading edge of the inner tongue (the part of the oligodendrocyte closest to the axon) and the lateral extension of myelin membrane layers towards the nodal region (the part of the oligodendrocyte the furthest from the axon)[44]. This growth is supported by a system of cytoplasmic channels providing shortcuts for the transport of membranes. Previous work established this process using the optic nerve of young mice (P10-60), which was dissected, HPF and then freeze substituted and stained. We were able to reproduce and extend this work in non-denaturing conditions in an adult mouse cortex by distinguishing and tracking thin and thick myelinated axons in 3 dimensions (Fig. 5A). Thin myelinated axons

have myelin sheaths of $19 \pm 4$ nm, while thick myelinated axon have a sheath of $40 \pm 11$ nm (Fig. 5 and Supplementary Fig. 10). The thin myelinated axon population is the only type for which we were to observe some axons along their entire length within our volume ($1134\,\mu m^3$) (Fig. 5 yellow box and in A, B and D). For this population, we also observed the network of cytoplasmic channels across a $1.6 \pm 1.3\,\mu m^3$ volume inside the myelin sheaths (Fig. 5A, D).

Within the thick myelinated axon population, axons with diameters both less than and greater than $1\,\mu m$ (Fig. 5 and Supplementary Fig. 10) were observed, which we as denote intermediate and large axons respectively. The intermediate axon population shows an extended network of cytoplasmic channels ($6.6 \pm 2.1\,\mu m^3$) to support the extended growth of the axon (Fig. 5A, B, E). On these axons, we also observed the progression from thin to thick myelin as the myelin compacts (Fig. 5E).

In some cases, it was possible to identify features in the data where the neurons changed morphology along their length. For example, we observed the start of a large axon (-1.5 $\mu m$ in diameter) (Fig. 5A, C, F) where, the cytoplasmic channels were not observed, as in other examples. Instead, breaks were visible within the myelin sheath associated with membranes from other cells, most probably oligodendrocytes, which are associated with additional membranes under the existing myelin. This suggests that large axons with thick myelin support myelination by the constant addition of oligodendrocytes tongues, in contrast to the mechanism of myelination in thin or intermediate axon populations.

The reduction of charging artefacts also allowed the visualisation of the immediate axon environment, identifying the extensive contacts that microglia have with the myelinated axon. Microglia cells play a crucial role in the maintenance of the myelin[63]. We observed that a single microglia cell has 3 extended regions of colocalization with 3 different axons (Fig. 5A, C, G). In close proximity, the plasma membrane of the microglia is sometimes impossible to distinguish from the myelin sheath given the image resolution but is consistent with their role a reported immunological survey of the CNS[64].

Using a modified scanning pattern, we observed and characterised different axons based on their diameter and myelin thickness which in turn supports different strategies for the formation and maintenance of the myelin sheath. The ability to observe different axonal phenotypes could be critical when mapping the different functions of axons within the brain, where some axons carry messages within the CNS while others at the periphery ensure anterograde or retrograde signalling. We also observed the interaction of microglia cells with different axons, demonstrating that the absence of charging artefacts allows for more features to be identified inside and outside the axons.

## Discussion

Volume imaging using FIB/SEM is a well-established technique used in both materials and life sciences[3,65–67]. However, most work reported in the literature describes results from samples at room temperature, which for many life science samples requires preparations that may introduce artefacts[68,69]. Furthermore, at cryogenic temperatures, the data can also be corrupted with curtaining or charging artefacts[20,28,29]. Here we have demonstrated that modifying the scanning pattern reduces the charging artefacts as illustrated for different exemplars of vitrified biological samples.

This paper focuses solely on an interleaved scan with a skip of 2 pixels in $x$ and $y$, in a frame size of $2048 \times 2048$ pixels. However, the relation between the interleaved steps and the image resolution requires further examination with regards to the probe size and spatial separation between scan positions. In principle, any number of pixels in the 2D array can be skipped while scanning in the $x$ and $y$ directions and alternative irregular scan geometries are also possible[34]. However, as the scan step increases, the response time of the scan deflectors may

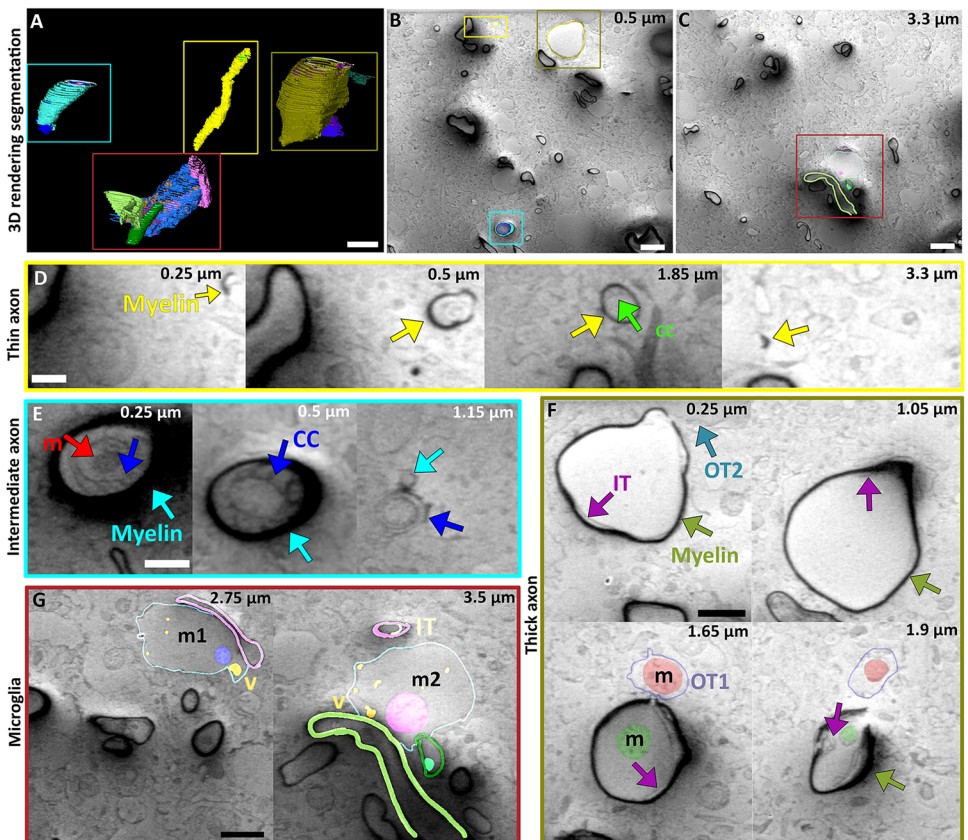

**Fig. 5 | SEM volume imaging of mouse brain. A** 118-day old mouse brain imaged at 90° with respect to the FIB milled sample plane using 100 ns dwell time ×100 repetitions. A 1334 μm³ volume in focus was aligned and manually segmented for the region of interest. **A** Segmented volumes. **B**, **C** Slices from the volume with overlayed segmentation. Scale bar 2 μm. Boxes in A are superimposed on B and C and enlarged in boxes (**D–G**) with the respective coloured box containing representative slices in the volume. Numbers indicate the z-location within the volume. **D** thin myelin axon with internal cytoplasmic channels (CC). Scale bar: 250 nm.

**E** intermediate axon with extended CC, mitochondria (m). Scale bar: 250 nm. **F** Thick myelin, large axon and the interaction with 2 oligodendrocytes showing the inner tongues (IT) and the outer tongues (OT). Scale bar: 500 nm. Mitochondria are also visible in the axon and one of the outer tongues. **G** Microglia cells and their proximity to different axons showing a stretch of colocalization, as well as internal organelles including mitochondria (m), vesicles (v) and inner tongues (IT). Scale bar 500 nm.

introduce errors in the probe positioning, leading to image distortions[31,70]. Additional distortions can arise from the imprinted scanning pattern on the acquisitions, a possible source could be residual charging artefacts modulating the contrast, resembling a checkerboard pattern; previously discussed in the context of the interplay between beam damage and the scanning sequence in STEM[26,31].

An interleaved pattern is adopted here to avoid any continuous changes of the scan direction for a constant scan speed and to simultaneously distribute the fluence more evenly. We have demonstrated that skipping a small number of pixels allows charge dissipation while maintaining low image distortion. However, we also observed that there is further scope to optimise the interleaved parameters as on some occasions charging artefacts were still present as halos surrounding a charging centre, which is particularly noticeable around LDs (Fig. 3F or I) or thick myelinated axons (Fig. 5F). Increasing the number of skipped pixels would potentially increase the scanning distortions due to drift and total scan time but would also reduce the elapsed time between visits of two neighbours which may not be sufficient for effective charge dissipation. Since the charge and discharge rate depends on the electrical resistivity and surface area of the sample[12], the optimal scan pattern is to a large extent dictated by the sample itself. Fundamentally, the scan step size is also linked to the magnification and the field-of-view, suggesting that individual optimisation may be necessary for specific experiments. A high magnification (and hence small pixel size) will require larger pixel number jumps to achieve the same sample distance as required for a given

charge dissipation rate. Finding an optimal pattern could be modelled by Monte Carlo simulations that take into consideration the spatial redistribution of charge as previously proposed[71,72]. Other optimisations based on a model for a diffusion mechanism could also be employed[34,73].

To separate the impact of interleaved scanning alone on the reduction of charging artefacts, other parameters (such as the accelerating voltage, imaging angle, beam current, pixel size, total fluence, working distance) and the detection system (in-lens secondary electron detector) have been deliberately restricted in this work. However, for example, image contrast has been improved by studying the effect of the accelerating voltage and the use of different detectors[16,74]. In addition, when imaging normal to the milled sample plane the energy crossover will change[75], and if not compensated this will introduce additional charging artefacts (Supplementary Figs. 4–6). Similarly, HPF samples have reduced charge dissipation as they form of a solid layer of non-conductive material and often require a decreased accelerating voltage for imaging[20,38]. However, the present approach has shown to be a robust method for charge dissipation and in future, the image quality may be further improved for specific samples by varying imaging parameters and by using different detectors.

In our experimental set up, we sometimes observed burn spots (approximately 100 nm size) that are not related to a specific biological structure but arise from beam damage. These are present when using the external scan engine and are more obvious during interleaved scanning (Fig. 3C, I and Supplementary Fig. 10E, F) due to the scan

being paused and the beam left unblanked between acquisitions. This pause occurs during the transfer of the analogue detector data from the scan unit to the host PC and also during the data type conversion, both of which occur in millisecond time scales. These burn spots can be mitigated by converting the data type after all frames are acquired and the information is saved, at the expense of increasing the amount of data on the host PC. An alternative option is to drive a beam blanker (with a shorter time response than the dwell time) by frame synchronization to blank the beam during the time the probe is paused before the start of the next frame.

As volumetric SEM imaging is used to track structures of interest, segmentation is frequently applied to the acquired images. To measure the impact of the reduction of charging artefacts on image interpretation, we applied the Segment Anything Model to our dataset. The presence of charging artefacts led to the incorrect identification of non-existent or distorted structures (Supplementary Fig. 8). Conversely, the absence of artefacts allowed more features to be selected with increased complexity. This improvement in segmentation due to the reduction of charging artefacts minimises human intervention, making on-the-fly segmentation and feature recognition for auto-mated SEM targeted milling feasible in the future.

To demonstrate the utility of our modified scanning approach, we have imaged several different relevant biological structures. LD, DC and axons were chosen as their compositions (mostly lipids and/or metabolites) and their insulating nature make them extremely sensitive to sample preparation and charging artefacts. Imaging these structures at high resolution is therefore at the limit of state-of-the-art SEM. Here we observed LD and DC environment and their interactions between different organelles. In *E. gracilis*, we were able to segment the network of DC, identify the different sub-populations, and we hypothesise that interactions between DC and other organelles, such as mitochondria and chloroplasts, lead to the emptying of the DC content. Better identification of features, resulting from further optimisation of the scanning pattern could inform additional correlative studies[20,29,30,76,77]. Finally, we have analysed two axons with different myelin thicknesses in adult mouse cortex. We imaged perpendicular to the milled plane of the sample to increase contrast[20] and due to the absence of artefacts were able to map specific axon phenotypes

In summary, we have demonstrated that by modifying the scanning pattern it is possible to substantially reduce charging artefacts, removing constraints on sample preparation for imaging specimens of varying conductivity. Combined with automated segmentation of features, volume SEM imaging with reduced artefacts provides a tool for automated recognition in FIB lamella preparation for in-situ structural biology for a future workflow.

## Methods
### Sample preparation
RPE-1 (CRL-4000) cells were obtained from the ATTC and cultured as per the supplier recommendation. For plunge freezing the cells were grown on EM grids (gold/200mesh/R2.2 either UltrAufoil or Quantifoil) before vitrification using a Vitrobot (ThermoFisher Scientific).

A 118-days-old (C57BL/6JH) mouse was euthanized using a schedule 1 procedure via intraperitoneal injection of sodium pentobarbital followed by decapitation following licensed procedures approved by the Mary Lyon Centre and the Home Office UK. All animals were handled according to approved and reviewed institutional animal care procedures. All mice (*Mus musculus*) were maintained and studied in accordance with UK Home Office legislation and local ethical guidelines issued by the Medical Research Council. Mice were fed *ad libitum* on a commercial diet (Teklad T2918MI 12) and had free access to water (reverse osmosis filtered and 9–13 ppm chlorinated). Mice were kept under controlled light (light 07:00–19:00, dark 19:00-07:00), temperature ($22 \pm 2 \,°C$) and humidity ($55 \pm 10\%$) conditions. All operating procedures were designed to minimise any suffering for the animals

involved in the study and were treated in accordance with the UK Animal (Scientific Procedures) Act 1986 Amendment Regulations 2012. The cortex was prepared as previously reported[20].

*Euglena gracilis* Klebs CCAP 1224/5Z was routinely cultured in *E. gracilis* medium and Jaworski's medium mixed in 1:1 proportion[78]. Cultures were maintained at 15 °C under a 12:12 h light: dark regime. Illumination was provided by cool white LED lamps with a photon flux density of 50 $\mu mol.m^{-2}.s^{-1}$ at the surface of the culture vessel. For plunge freezing the cells were pipetted (4 μl) on EM grids (gold/200mesh/R2.2 either UltrAufoil or Quantifoil) in a humidified chamber (98% RH), single side back blotted (20 s) before vitrification by plunging into liquid ethane using an EM GP Plunge Freezer (Leica microsystems, Austria).

### SEM imaging
An external scan engine (Quantum Detectors) was interfaced to a Helios Hydra G5 (ThermoFisher Scientific) using the available external scan control of the microscope. The cryo-stage and the anti-contamination strategy is adapted from the Aquilos II system (ThermoFisher Scientific). The scan inputs to control the SEM deflectors were controlled by the Quantum Detector scan engine and the signal from the scintillator/photomultiplier secondary electron detector was connected to one of its analog inputs. Supplementary Fig. 11 illustrates the appropriate connections between the microscope and the scan engine used in this work. The microscope was operated in external scan mode to allow control of the deflector with the external scan engine. The microscope allows access to this mode either by an option on the user interface or by scripting using AutoScript (ThermoFisher Python-based scripting software). The scan engine software was based on a Python module with the data saved as 2D NumPy arrays, as int16 data type by default. The pattern for the interleaved scanning was generated using an in-house Python script (see data and code availability) and skips two pixels in the *x* scanning direction and two lines in the *y* scanning direction, as illustrated in Fig. 1. This script provides a text file of *x* and *y* coordinates, which is read by the external scan generator to control the position of the electron probe for imaging. The control is only used for the SEM probe deflectors. AutoScript was used to integrate SEM imaging using the programmable scan engine during FIB/SEM volumetric imaging. To limit the number of variables in our experiments, we defined the base parameters to match our previous work[20,38]. In summary, the parameters used for SEM imaging were accelerating voltage 1.2 kV, beam current 6.3 pA, probe size approximately 2 nm at FWHM, dwell time 100 - 1000 ns, 10 – 100 repetitions, immersion mode (suction tube 70 eV), detector; in-lens for secondary electrons, pixel size 6.34 nm. The detector voltage offset and gain were maintained per dataset for each sample but not between samples. Flyback delay during acquisitions was not applied when using frame integration for both raster and interleaved patterns, to ensure that all images were acquired with the same electron fluence. This flyback delay is commonly applied at the start of each raster scan line to wait for stabilisation of the probe after it is moved from the end of one line to the next one[31,70]. However, flyback delay was applied to raster line integration acquisitions as these were acquired using the inbuilt functionality of the microscope (accessed through the microscope's user interface) which does not allow deactivation of the flyback delay (of approximately 70 μs for the conditions used in our experiments). Beam damage or charging effects were not observed on the left hand-side of line integration acquisitions since the probe is only stationary for a very short time. The acquisitions were cropped to perform the analysis on images without distortions, as described below. For consistency in comparisons, the raster line integration images were also cropped.

### Automation with SerialFIB
As for previous implementations of customised SEM imaging procedures[20,38], the scanning strategies reported here were integrated to the software package SerialFIB[79]. The volume imaging script developed for different milling and image acquisition positions that enables

perpendicular SEM imaging was extended using the scripts above to control the scanning patterns. To compensate an issue when switching to SEM scanning after FIB milling, a preview image at a dwell time of 25 ns was acquired prior to the interleaved scanning exposure. Data acquisition was implemented using the customised scripts described above, writing out single frames per interleaved scan after acquisition.

## Pre-processing−estimation of the spatial extent of flyback distortions

The absence of a flyback delay leads to distortions and then poor image quality on the left side of the acquisitions that extend to a number of pixels. For this reason, to remove distortions, raster and interleaved acquisitions were cropped. This number of pixels was estimated from the interleaved images acquired using the shortest dwell time since these settings will result in maximum distortions. This number have been applied to all images both raster and interleaved.

In more detail, we implemented an automated analysis (data and code availability section), to statistically estimate this factor from a small region in each frame of a sequential acquisition that shows obvious flyback distortions. The same region is selected for all frames in the experiment, as shown in Supplementary Fig. 12. A Gaussian filter, $\sigma = 7$ pixels, was applied to increase the SNR and subsequently a Sobel filter ($3 \times 3$ kernel), for gradient approximation and a threshold for edge detection to convert the grey-scale images into binary images. The result is a vertical band whose right edge (rightmost pixel with intensity value one) represents the spatial extent of the flyback distortions. This approach was applied to the RPE-1 dataset acquired using 100 ns dwell time and 100 repetitions. A histogram was calculated from the measurements of all the acquisitions, and the mean value of the normal distribution taken as a reference to crop the images acquired from different samples. The histogram indicates 64 pixels as the extent of the distortions. We considered this value a valid approximation since the Gaussian filter smooths fine details, inherently extending the edge of the apparent flyback distortions. The outliers in the histogram are the result of the edge detection process failing to find clear edges in a noisy background.

## Subframe preparation

Since a relatively large amount of data was collected, particularly during volumetric imaging, the 2D Numpy arrays were converted to a uint8 data type during acquisition and stored as raw data; given that grayscale levels in 8-bit images are not visually distinguishable and are adequate for the image processing performed in this work[80]. The conversion was done by shifting the histogram of the acquisitions to a positive range and subsequently rescaling the pixel intensities. Immediately after acquisition, the 2D Numpy arrays were then grouped and stacked to create a 3D array and saved in a tiff format (see data and code availability section). Cropping was performed post-acquisition to remove flyback artefacts. For frame integration, the frames were aligned using MotionCor2[27], with a $5 \times 5$ patch and 20% overlap. For volumetric data, the frames (or images) were aligned using a SIFT[81] plugin in Fiji[82] using the default parameters for the affine option without interpolation[81].

## Histogram analysis

The mean value of the histograms of aligned subframes were calculated from the cropped data ($1920 \times 1920$ pixels) using a script for batch processing in Fiji. Two tails T-test alpha 5%. At least, 5 samples were measured per imaging conditions irrespective of the sample type allowing for statistical relevance while maintaining feasibility.

## Myelin sheath thickness measurement

For the volume of the brain acquired, lines were manually placed on the myelin sheath where intense dark pixels were present. We used 10 measurements over 3 independent axons and measured their length using the Fiji build-in measurement tool. T-tests were performed over the two populations that show that they are independent with a confidence of 1%. At least, 3 axons were measured per conditions allowing for statistical relevance while maintaining feasibility.

## Segmentation and subsequent data analysis

Aligned and cropped subframes, forming a frame or image, were filtered using a Fiji mean filter with a $3 \times 3$ filter kernel.

## Unsupervised SAM segmentation

Images were loaded using a Python script and processed by the *SamAutomaticMaskGenerator* from Segment Anything[54]. The Segment Anything model used was *ViT-H* with 32 points_per_side, a value of 0.92 for *pred_iou_thresh* and 0.9 for *stability_score_thresh* with other parameters set to default values. The perimeter and area of each segmented component was calculated with *skimage regionprops* and a complexity score was calculated as the squared perimeter of the component divided by the area. These scores were summed for all components in the segmentation to produce a final image complexity score. Statistical tests were performed to evaluate the difference between a population and a set value (in this case 1) that show that they are independent with a confidence of 5%. At least 4 samples representing all the cell/tissue type described in this study were measured and this per imaging conditions allowing for statistical relevance while maintaining feasibility.

## Manual segmentation

For *E. gracilis*, in addition to the mean filter, the *remove background option*[83] in Fiji was used with a rolling ball of 50 pixels and the light background option.

Amira (ThermoFisher Scientific) was used to manually segment volumes which were then exported as a tiff stacks for further analysis using Fiji. BioVoxxel 3D, Image 3D suite and their dependencies were used for volume calculations of the different segmented objects[80,84,85]. A quadrant count ratio was performed on the entire population of vacuoles (26) in one *E. gracilis* cell as we were interested in a per-cell analysis.

## Reporting summary

Further information on research design is available in the Nature Portfolio Reporting Summary linked to this article.

# Data availability

The volume acquired for *E. gracilis* and adult mouse cortex, as well as the segmented volumes, are available on EMPIAR with access codes 12244 and 12239. Source data are provided with this paper.

# Code availability

Scripts for data acquisition and pre-processing are available on the Rosalind Franklin Github (https://github.com/rosalindfranklininstitute/fib-sem-charge-mitigation)[86]. For SerialFIB the software and specific scripts for the scan engine are available on GitHub (https://github.com/sklumpe/SerialFIB).

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

## Acknowledgements

The authors would like to acknowledge the help of Matthew Case for analysis of the brain samples. Kamlesh Patel from ThermoFisher Scientific, Leo Jay-Black and Ben Bradnick from Quantum Detectors for the integration of the scan engine/microscope and for the Python module, respectively, and Elaine Ho for her help with Motioncor2. The authors would also like to thank the experimental hall coordinators at Diamond Light Source for their monitoring of our system 24/7 and the Mary Lyon Centre for their excellent care to animals. This work was supported by the Wellcome Trust through the Electrifying Life Science grant (220526/Z/20/Z to J.H.N.) and a Wellcome Trust Development Award (225902/Z/22/Z to M.G.). The Rosalind Franklin Institute is funded by the UK Research and Innovation, Engineering and Physical Sciences Research Council.

## Author contributions

A.V. conceptualised, developed the acquisition protocol and acquired data. T.G. integrated the scan engine with the microscope acquisition process and acquired data. S.K. automatized the data acquisition protocol. A.P. performed the automated segmentation. J.Z. performed manual segmentation. J.L.R.S., C.G. and M.K. prepared the samples. W.B. performed preliminary data acquisition experiments. R.A.F., J.H.N., J.S.K., M.C.D., M.G., A.I.K. and M.D. supervised the work. M.D. also performed manual segmentation and interpreted the data. A.V., M.D. and

J.S.K. conceptualised the initial method development. All authors commented on and edited the manuscript.

## Competing interests

The authors declare no competing interests.
