## [Transparent Peer Review file · Nature Communications]

Reduction of SEM charging artefacts in native cryogenic biological samples

Corresponding Author: Dr Maud Dumoux

Version 0:

Reviewer comments:

Reviewer #1

(Remarks to the Author)

The paper describes a novel method of scanning the electron beam in a manner that reduces charging. The authors demonstrate the effectiveness of their approach by imaging cryo-SEM specimens of various types of cells. I believe that it should be published in Nature Communications after taking into account several comments that I list below.

The authors are well aware of the important effects of the acceleration voltage on specimen charging, and the possibility to use the optimal voltage that leaves a non-conductive specimen neutral, thus avoiding charging effects. Indeed, they refer to the important seminal work of Joy and others. That leads me to two questions. First the authors state that they used an acceleration voltage of about 1 kV. They don't specify it more accurately, and seem to not to have tried to optimize it. Why? They also disregard the effect of the acceleration voltage on the micrograph contrast. I would like to point out to them a paper published some years ago that deals with that effect (Lieberman, L., Kleinerman, O., Davidovich, I., and Talmon, Y., Micrograph Contrast in Low-Voltage SEM and Cryo-SEM, Ultramicroscopy 218, 113085 (2020)). That paper also points out that when using a backscattered electron detector (the authors reported using only a secondary electron detector) one can obtain very good contrast between water-rich and oil-rich domains, which could be quite useful in the study biological specimens.

An additional issue is how easy or difficult is it to apply this demonstrated mode of scanning the beam in standard commercial scanning electron microscopes.

Reviewer #2

(Remarks to the Author)

Scanning electron microscopy of insulating material meets some obstacles of which charging is the biggest problem. Velazco et al describe a new scanning regime that allows the electrons more time to dissipate during image acquisition, resulting in a considerable reduction of charging. The new regime has been applied on 3 different samples prepared for cryo-FIB volume microscopy.

The presented data is convincing that the new method dramatically decreases charging artefacts and improves image quality. This way also automated segmentation is improved, which is essential for analysing the huge amount of data generated by volume microscopy.

In this manuscript my concern goes out to the interpretation of the images. In my opinion the authors read more details from the images than the resolution justifies.

They state that the voxel resolution is 6.34 nm x 6.34 nm x 50 nm, which does not allow analysing membrane – membrane interactions. The membrane thickness is about 5 nm. To securely analyse membrane – membrane interactions we must see the individual leaflets of the membrane.

At a resolution of ~13 nm (6.34 nm pixel size) membrane contacts and membrane hemi-fusion (Fig 3 I and discussion 506-508) are not resolved. For both the resolution needs to be in the range of 1 nm. In the discussion (473 – 476) the authors state: '...as on some occasions charging artefacts appearing as halo surrounding a charging centre, which is particularly

noticeable around LDs (Figure 3F or I) or thick myelinated axons (Figure 5F'), which makes their optimistic interpretation even less substantiated.

The same is true for the contact sites of *C. gracilis* (Fig 4 and 508-512) and the contacts described in the mouse brain, Fig 5. The charging at the axons, though very much reduced, is still too prominent and the resolution too limited to visualise membrane-membrane contacts.

In Fig 4C, E and Fig 5F, G the mitochondria (m) are blurred and their identification is rather guessing.

The difference between the thickness of the myelin sheet of the thick and thin axons is large enough to be visualised by, e.g., by cryo-fixation / freeze-substitution, at higher resolution without considerable artefacts. Therefore I suggest deleting the sentence: 'These different phenotypes could not be identified at room temperature due to sample preparation and/or imaging artefacts' (515,516)

The burn spots mentioned (487-491) should be indicated on the Fig 3C, I and Supplementary Figure 10E, F. Here the question arises, what is the total electron dose per Å².

My critics concern the exaggerated interpretation of the cryo-EM data. The interpretation must be realistic based on the resolution of the images. This part needs rewriting!

The new scanning regime is extremely promising and as the authors state there is improvement potential by adapting the scanning matrix, e.g., using 3 by 3 instead of 2 by 2 steps.

In my opinion this method will have a much greater impact on improving imaging with serial block face scanning microscopy than cryo-FIB.

I strongly support the publication of this manuscript in Nature Communications, after the data interpretation has been revised to be realistic for the resolution of the images.

Reviewer #3

(Remarks to the Author)

This paper addresses the problem of charging, that occurs everytime one performs scanning electron imaging on poorly conductive samples (a category where biological samples fall). These artifacts particularly affect insulating areas of cryogenically preserved samples, limiting the ability to observe important biological features in their native state.

The authors present a very nice and simple solution by introducing an interleaved scanning pattern, which skips pixels during scanning rather than following the conventional raster pattern. This approach allows more time for charge dissipation between scanned points, significantly reducing charging artifacts. They demonstrated their method's effectiveness using three different biological samples such as cultured mammalian cells, single-cell algae (*E. gracilis*), and mouse brain tissue. This approach was already used in STEM and FIB milling and seeing it applied to cryoSEM imaging is an excellent addition to the field.

The paper represents an important methodological advancement in cryo-electron microscopy, offering a practical solution to a long-standing impediment. One point I particularly appreciate is that the authors are transparent about the limitations of their work, particularly in the technical implementation (e.g. burn spot issues).

There are a few points that I would like to make to improve this work.

1. Technical Parameters and Optimization:

The paper only explores a 3x3 interleaved scan pattern but doesn't fully explain why this specific pattern was chosen. I understand that the idea was to leave some space between the individual exposures. But curiously, the authors explore the effect of different dwell times but have put no emphasis on the scanning patterns.

Also, I found a very limited explanation of how the optimal number of pixels to skip should be determined for different sample types.

Finally on this point I would appreciate more discussion of how varying the scan pattern parameters might affect image quality on different types of samples.

2. Methodological Limitations:

The authors acknowledge but don't fully address the "burn spots" issue that occurs during interleaved scanning. It would be helpful to have a short discussion on how to deal with this problem, which I understand is likely due to the current hardware setup.

Further, I would find it very informative to have a better comparison with other existing charge mitigation techniques like variable pressure or gas injection systems. I understand that this is probably not achievable in their system, but a more extensive discussion on this point would be helpful.

3. Data Analysis:

The statistical significance of the improvements in image quality could be better quantified and I would attempt to see if metrics for determining optimal imaging conditions for different sample types could be available. For example the authors

may use a FRC to quantify the image stability and improvement in relation to different amount of charging. similarly to what is commonly done in super-resolution microscopy.

4. Technical Implementation:

The integration of the external scan engine with existing microscope systems could be explained in more detail, The authors may provide provide better discussion in regards to compatibility issues with different microscope manufacturers and a clear guidance on how to implement this technique in laboratories with different equipment setups.

Most of the requests I made can be handled with text improvement and are only aimed at improving the overall output, The only two real experiments/analyses I am asking for are: the FRC or any other quantification of the image quality and stability and testing different scanning patterns, whose benefit can be in principle directly quantified with the FRC.

Happy to discuss this in further detail if required.

Alex de Marco

Version 1:

Reviewer comments:

Reviewer #1

(Remarks to the Author)

The authors modified their manuscript according to my suggestions. I am pleased with the manuscript in its present form, and recommend it is published in Nature Communications.

Reviewer #2

(Remarks to the Author)

The manuscript has greatly improved.

Still I cannot SEE MCS nor mitochondria on the images of the manuscript I have at hand. Mitochondria are expected and most likely but cannot clearly be addressed at this resolution. The MCS cannot be resolved at the resolution used in this research, in fact they can only be identified at resolutions achieved by cryoTEMs.

Reviewer #3

(Remarks to the Author)

I am happy with the revision,
nice work.

Alex

We thank the reviewers for their helpful comments on this work which have enabled us to improve the manuscript. Our responses to specific comments are given below and changes in the manuscript text are reproduced in red and highlighted in yellow in the main manuscript.

Text colour key:

Reviewer comments

Author responses

Modified manuscript text

Reviewer #1 (Remarks to the Author):

The paper describes a novel method of scanning the electron beam in a manner that reduces chargining. The authors demonstrate the effectiveness of their approach by imaging cryo-SEM specimens of various types of cells. I believe that it should be published in Nature Communications after taking into account several comments that I list below.

The authors are well aware of the important effects of the acceleration voltage on specimen charging, and the possibility to use the optimal voltage that leaves a non-conductive specimen neutral, thus avoiding charging effects. Indeed, they refer to the important seminal work of Joy and others. That leads me to two questions. First the authors state that they used an acceleration voltage of about 1 kV. They don't specify it more accurately, and seem to not have tried to optimize it. Why? They also disregard the effect of the acceleration voltage on the micrograph contrast. I would like to point out to them a paper published some years ago that deals with that effect (Lieberman, L., Kleinerman, O., Davidovich, I., and Talmon, Y., Micrograph Contrast in Low-Voltage SEM and Cryo-SEM, Ultramicroscopy 218, 113085 (2020)). That paper also points out that when using a backscattered electron detector (the authors reported using only a secondary electron detector) one can obtain very good contrast between water-rich and oil-rich domains, which could be quite useful in the study of biological specimens.

We have added the exact accelerating voltage of 1.2kV for biological samples taken from our previous work. We have also modified the text to emphasise this point and added the suggested reference to Lieberman's work (citation 16).

The revised text is reproduced below.

"In our previous work^{20,38} we demonstrated that using an Ar plasma for sputtering followed by secondary electron imaging at low kV (1.2 kV) it was possible to image volumes of vitrified biological samples at ~10 nm resolution. In the present study we define an accelerating voltage and set a constant electron fluence ($10^{-1} \text{ e}^-/\text{\AA}^2$). This electron fluence was distributed using three combinations of dwell time and number of scan repetitions to vary the pixel flux in three regimes as: low (100 ns x100 repetitions), intermediate (500 ns x20 repetitions) and high (1000 ns x10 repetitions). The repetitions were completed either at the line integration (LI) or frame integration (FI) level and the scan pattern was either raster or interleaved."

We have also corrected a typo. The fluence is corrected from 10^{-2} to $10^{-1} \text{ e}^-/\text{\AA}^2$.

In addition, we have also more clearly stated in our conclusions that optimisation of kV and detector signals was intentionally omitted to focus on the scanning effect, but it would be an excellent next study and as Reviewer #1 notes it could vary based on sample composition.

The revised text is reproduced below.

“To separate the impact of interleaved scanning alone on the reduction of charging artefacts, other parameters (such as the accelerating voltage, imaging angle, beam current, pixel size, total fluence, working distance) and the detection system (in-lens secondary electron detector) have been deliberately restricted in this work. However, for example, image contrast has been improved by studying the effect of the accelerating voltage and the use of different detectors^{16,75}. In addition, when imaging normal to the milled sample plane the energy crossover will change⁷⁶, and if not compensated this will introduce additional charging artefacts (Supplementary Figures 4-6). Similarly, HPF samples have reduced charge dissipation as they form of a solid layer of non-conductive material and often require a decreased accelerating voltage for imaging^{20,38}. However, the present approach has shown to be a robust method for charge dissipation and in future, the image quality may be further improved for specific samples by varying imaging parameters and by using different detectors.”

An additional issue is how easy or difficult is it to apply this demonstrated mode of scanning the beam in standard commercial scanning electron microscopes.

We have modified the Materials and Methods section and added a supplementary figure (now Supplementary Figure 11) to provide more details on the installation of the scan engine on our system. Overall, it took approximately 96 hours to have the system up and running for single image acquisitions and an additional 48 hours for integration with automated image acquisitions software packages (ie SerialFIB or automated slice and view).

A similar comment has been made by Reviewer #3.

The revised text is reproduced below

“An external scan engine (Quantum Detectors) was interfaced to a Helios Hydra G5 (ThermoFisher Scientific) using the available external scan control of the microscope. The cryo-stage and the anti-contamination strategy is adapted from the Aquilos II system (ThermoFisher Scientific). The scan inputs to control the SEM deflectors were controlled by the Quantum Detector scan engine and the signal from the scintillator/photomultiplier secondary electron detector was connected to one of its analog inputs. Supplementary Figure 11 illustrates the appropriate connections between the microscope and the scan engine used in this work. The microscope was operated in “external scan” mode to allow control of the deflector with the external scan engine. The microscope allows access to this mode either by an option on the user interface or by scripting using AutoScript (ThermoFisher Python-based scripting software). The scan engine software was based on a Python module with the data saved as 2D NumPy arrays, as int16 data type by default. The pattern for the interleaved scanning was generated using an in-house Python script (see data and code availability) and skips two pixels in the x scanning direction and two lines in the y scanning direction, as illustrated in Figure 1. This script provides a text file of x and y coordinates, which is read by the external scan generator to control the position of the electron probe for imaging. The control is only used for the SEM probe deflectors.

AutoScript was used to integrate SEM imaging using the programmable scan engine during FIB/SEM volumetric imaging.”

Reviewer #2 (Remarks to the Author):

Scanning electron microscopy of insulating material meets some obstacles of which charging is the biggest problem. Velazco et al describe a new scanning regime that allows the electrons more time to dissipate during image acquisition, resulting in a considerable reduction of charging. The new regime has been applied on 3 different samples prepared for cryo-FIB volume microscopy.

The presented data is convincing that the new method dramatically decreases charging artefacts and improves image quality. This way also automated segmentation is improved, which is essential for analysing the huge amount of data generated by volume microscopy.

In this manuscript my concern goes out to the interpretation of the images. In my opinion the authors read more details from the images than the resolution justifies.

They state that the voxel resolution is 6.34 nm x 6.34 nm x 50 nm, which does not allow analysing membrane – membrane interactions. The membrane thickness is about 5 nm. To securely analyse membrane – membrane interactions we must see the individual leaflets of the membrane. At a resolution of ~13 nm (6.34 nm pixel size) membrane contacts and membrane hemi-fusion (Fig 3 I and discussion 506-508) are not resolved. For both the resolution needs to be in the range of 1 nm. In the discussion (473 – 476) the authors state: ‘...as on some occasions charging artefacts appearing as halo surrounding a charging centre, which is particularly noticeable around LDs (Figure 3F or I) or thick myelinated axons (Figure 5F’), which makes their optimistic interpretation even less substantiated.

The same is true for the contact sites of C. gracilis (Fig 4 and 508-512) and the contacts described in the mouse brain, Fig 5.

The charging at the axons, though very much reduced, is still too prominent and the resolution too limited to visualise membrane-membrane contacts.

In Fig 4C, E and Fig 5F, G the mitochondria (m) are blurred and there identification is rather guessing.

The difference between the thickness of the myelin sheet of the thick and thin axons is large enough to be visualised by, e.g., by cryo-fixation / freeze-substitution, at higher resolution without considerable artefacts. Therefore I suggest deleting the sentence: ‘These different phenotypes could not be identified at room temperature due to sample preparation and/or imaging artefacts’ (515,516)

We have now removed the statements regarding the membrane contact sites (MCS and have modified the main text, figure annotations and legend accordingly to describe the biological event more accurately as follows:

1) For the mammalian cells: we have modified the text (reproduced below), Figure 3 and its figure caption.

“The additional information in images formed using interleaved scanning allows the observation of endoplasmic reticulum (ER) wrapped around LD (Figure 3G-I), suggesting a MCS, demonstrating that

the alternative scanning strategy has a positive impact on studies of the interaction of LD with other organelles in native conditions.”

Figure 3 New caption:

Figure 3: Charging artefact reduction in RPE-1 cells allows the observation of lipid droplet (LD) surrounding environments and LD to organelle membrane contact sites (MCS).

Vitrified cells imaged at 52° with respect to the FIB milled sample plane using a 100 ns dwell time x100 repetitions. (A-C) Overview of a slice from RPE-1 imaged using raster line integration, raster frame integration and interleaved frame integration, respectively. Scale bar: 2 µm. (D-F) enlargement of the pink boxes in (A to C). White arrow: membrane in the vicinity of a LD. Black arrow: content of a degradative compartment. m: mitochondria. Scale bar 0.5 µm. (G-I) Enlargement of the green boxes in (A to C); brightness and contrast were modified to improve visualisation. White arrow: wrapping of endoplasmic reticulum (ER) around LD. Scale bar 1 µm.

2) For *E.gracilis* : we have modified the text (reproduced below), and the caption to Figure 4 (also reproduced below).

“We have also observed the environment of these different populations and noticed the presence of extended contact between the nucleus, mitochondria or chloroplasts and the DC, but are not yet able to differentiate between the two membranes of the organelles (Figure 4C and D).

This hypothesis is supported by the observation of emptied vesicles in close proximity to DC (Figure 4E) and we propose that these empty vesicles are emptied degradative compartments.”

Figure 4 New Caption:

Figure 4: SEM volume imaging of *E.gracilis*.

E.gracilis imaged at 52° with respect to the FIB milled sample plane using 100 ns dwell time x100 repetitions. A 1600 µm³ volume in focus aligned and subject to manual segmentation of the region of interest. (A, C-G) Images after background removal (see material and methods) to assist segmentation. For (A, C-G) the number in the upper corner is the z-location within the volume. (A) Membranes of the degradation compartment segmented (red) and content (blue). Scale bar: 2 µm. (B) Density (ratio of the content volume within the degradative compartment) as a function of the volume of the degradative compartment showing the presence of four unrelated populations (quadrant count ratio = 0.037). (C-E) Enlarged panels from the coloured boxes in (A), highlighting the membrane deformation of organelles in close proximity to the degradation compartment (DC) or empty vesicles and other organelles including mitochondria (m) and chloroplast (C) or an absence of contact with, for example, the Golgi apparatus (GA). The limits of these contacts are indicated by a black arrow. Scale bars: 500 nm (C), 250 nm (D-E). (F-G) Enlarged panels of the eyespot and 3D segmentation, proximal to the reservoir (R). Compartments in proximity (V: vesicle, C: chloroplast). Scale bar: 250 nm.

3) Mouse brain data: we have modified the text (reproduced below), and amended part of the caption to Figure 5 (also reproduced below).

“We observed that a single microglia cell has 3 extended regions of colocalisation with 3 different axons (Figure 5A, C and G). In close proximity, the plasma membrane of the microglia is sometimes impossible to distinguish from the myelin sheath given the image resolution, but is consistent with their role a reported immunological survey of the CNS⁶⁵.”

Figure 5 Amended part of caption:

(F) thick myelin, large axon and the contact interaction with 2 oligodendrocytes showing the inner tongues (IT) and the outer tongues (OT). Scale bar: 500 nm. Mitochondria are also visible in the axon and one of the outer tongues. (G) microglia cells and their proximity with different axons showing a stretch of colocalisation, as well as internal organelles including mitochondria (m), vesicles (v) and inner tongues (IT). Scale bar 500 nm.

The Conclusions section has also been modified to reflect the above changes, as reproduced below.

“Here we observed LD and DC environment and their interactions between different organelles. In *E.gracilis*, we were able to segment the network of DC, identify the different sub-populations and we hypothesise that interactions between DC and other organelles, such as mitochondria and chloroplasts, lead to the emptying of the DC content. Better identification of features, resulting from further optimisation of the scanning pattern could inform additional correlative studies^{20,29,30,77,78}. Finally, we have analysed two axons with different myelin thicknesses in adult mouse cortex. We imaged perpendicular to the milled plane of the sample to increase contrast²⁰ and due to the absence of artefacts were able to map specific axon phenotypes. “

The burn spots mentioned (487-491) should be indicated on the Fig 3C, I and Supplementary Figure 10E, F. Here the question arises, what is the total electron dose per Å².

My critics concern the exaggerated interpretation of the cryo-EM data. The interpretation must be realistic based on the resolution of the images. This part needs rewriting!

The new scanning regime is extremely promising and as the authors state there is improvement potential by adapting the scanning matrix, e.g., using 3 by 3 instead of 2 by 2 steps.

Regarding the burn spots, Supplementary Figure 10E and F and corresponding legend have been modified to indicate their presence. The electron fluence for the entire image is $10^{-1} \text{ e}/\text{Å}^2$. However, the formation of the burn spots is not controllable in our current instrument, and we do not have an accurate measurement of the time it requires to form and therefore cannot calculate the fluence required to generate the burn spots. We have added a discussion about the options to avoid such burn spot in future, with the revised text reproduced below.

“In our experimental set up, we sometimes observed ‘burn spots’ (approximately 100 nm size) that are not related to a specific biological structure but arise from beam damage. These are present when using an external scan engine and are more obvious during interleaved scanning (Figure 3C, I and Supplementary Figure 10E, F) due to the scan being paused and the beam left unblanked between acquisitions. This pause occurs during the transfer of the analog detector data from the scan unit to the host PC and also during the data type conversion, both of which occur in millisecond time scales. These burn spots can be mitigated by converting the data type after all frames are acquired and the information is saved, at the expense of increasing the amount of data on the host PC. An alternative option is to drive a beam blanker (with a shorter time response than the dwell time) by frame synchronization to blank the beam during the time the probe is paused before the start of the next frame.”

In order to keep the fluidity of the argument developed in the manuscript, we did not modified figure 3 but instead made an enlarged version of panel C of figure 3 in this response document, with the burn spots highlighted in yellow.

Response Figure 1: Enlargement of figure 3C, in yellow the burn spots. Scale bar: 2 μ m.

In my opinion this method will have a much greater impact on improving imaging with serial block face scanning microscopy than cryo-FIB.

We have now mentioned serial block face in in the main text bas suggested.

I strongly support the publication of this manuscript in Nature Communications, after the data interpretation has been revised to be realistic for the resolution of the images.

We have, as suggested limited some of our more speculative interpretations as suggested.

Reviewer #3 (Remarks to the Author):

This paper addresses the problem of charging, that occurs everytime one performs scanning electron imaging on poorly conductive samples (a category where biological samples fall). These artifacts particularly affect insulating areas of cryogenically preserved samples, limiting the ability to observe important biological features in their native state.

*The authors present a very nice and simple solution by introducing an interleaved scanning pattern, which skips pixels during scanning rather than following the conventional raster pattern. This approach allows more time for charge dissipation between scanned points, significantly reducing charging artifacts. They demonstrated their method's effectiveness using three different biological samples such as cultured mammalian cells, single-cell algae (*E. gracilis*), and mouse brain tissue. This approach was already used in STEM and FIB milling and seeing it applied to cryoSEM imaging is an excellent addition to the field.*

The paper represents an important methodological advancement in cryo-electron microscopy, offering a practical solution to a long-standing impediment. One point I particularly appreciate is that the authors are transparent about the limitations of their work, particularly in the technical implementation (e.g. burn spot issues).

There are a few points that I would like to make to improve this work.

1. Technical Parameters and Optimization:

The paper only explores a 3x3 interleaved scan pattern but doesn't fully explain why this specific pattern was chosen. I understand that the idea was to leave some space between the individual exposures. But curiously, the authors explore the effect of different dwell times but have put no emphasis on the scanning patterns.

Also, I found a very limited explanation of how the optimal number of pixels to skip should be determined for different sample types.

Finally on this point I would appreciate more discussion of how varying the scan pattern parameters might affect image quality on different types of samples.

We did not present data from the exploration of the impact of the scanning pattern on the charging artefact as it constitutes a wide search space including number of pixels jumped but also geometry of the scanning sequence (linear, serpentine, Hilbert, etc.). Fundamentally, this is linked to the image magnification which determines the spacing of the probe positions and should be explored based on the balance between experimentally needed resolution and field-of-view. It is likely that higher magnifications would require larger pixel number jumps in the same material to account for the rate of charge dissipation. Another consideration is the impact of the modification of the scanning pattern on other parameters such as the size of the flyback and the image distortion (see Figure 1 below) both of which could be corrected by modifying additional parameters, such as flyback time, or modelling the effects of these distortions for bespoke post-processing. .

Regarding the choice of the 3x3 pattern we now have added the following text in the Results section, with the revised text reproduced below.

“A linear pattern has been chosen as previous work demonstrates good performance with regards to image distortion in scanning-TEM (STEM)³¹ while the skipping of 2 pixels in x and y has demonstrated damage reduction in high resolution STEM imaging²⁶.”

Response Figure 2: effect of the number of skipping steps on charge mitigation and image distortion. *E.gracilis* imaged using 1.2 kV, 6.25 pA, 100 ns x100 repeats using either raster scanning combined with line integration or interleaved scanning skipping different number of pixels and combined with frame integration. Skip 2 is the equivalent of the 3x3 scanning pattern described in the text. Top row: 3 μm. Bottom row: 1 μm. Top row highlights the increased size of distortion on the left of the image, illustrating the error on the landing position of the beam after the flyback. Bottom row is a zoom from the top panel to highlight the presence of charging artifact or image distortion as a blurring of the image.

We have also added some further discussion on this point in the Conclusions section, with the revised text reproduced below

“However, as the scan step increases, the response time of the scan deflectors may introduce errors in the probe positioning, leading to image distortions^{31,71}. Additional distortions can arise from the imprinted scanning pattern on the acquisitions, a possible source could be residual charging artefacts modulating the contrast, resembling a checkerboard pattern; previously discussed in the context of the interplay between beam damage and the scanning sequence in STEM^{26,31}.

An interleaved pattern is adopted here to avoid any continuous changes of the scan direction for a constant scan speed and to simultaneously distribute the fluence more evenly. We have demonstrated that skipping a small number of pixels allows charge dissipation while maintaining low image distortion. However, we also observed that there is further scope to optimise the interleaved parameters as on some occasions charging artefacts were still present as halos surrounding a charging centre, which is particularly noticeable around LDs (Figure 3F or I) or thick myelinated axons (Figure 5F). Increasing the number of skipped pixels would potentially increase the scanning distortions due to drift and total scan time but would also reduce the elapsed time between visits of two neighbours which may not be sufficient for effective charge dissipation. Since the charge and discharge rate depends on the electrical resistivity and surface area of the sample¹², the optimal scan pattern is to a large extent dictated by the sample itself. Fundamentally, the scan step size is also linked to the magnification and the field-of-view, suggesting that individual optimisation may be necessary for specific experiments. A high magnification (and hence small pixel size) will require larger pixel number jumps to achieve the same sample distance as required for a given charge dissipation rate. Finding an optimal pattern could be modelled by Monte Carlo simulations that take into consideration the spatial

re-distribution of charge as previously proposed^{72,73}. Other optimisations based on a model for a diffusion mechanism could also be employed^{34,74}.”

2. Methodological Limitations:

The authors acknowledge but don't fully address the "burn spots" issue that occurs during interleaved scanning. It would be helpful to have a short discussion on how to deal with this problem, which I understand is likely due to the current hardware setup.

We agree with the reviewer and have extended the discussion on this topic in the Conclusions section, with the revised text reproduced below

“In our experimental set up, we sometimes observed ‘burn spots’ (approximately 100 nm size) that are not related to a specific biological structure but arise from beam damage. These are present when using the external scan engine and are more obvious during interleaved scanning (Figure 3C, I and Supplementary Figure 10E, F) due to the scan being paused and the beam left unblanked between acquisitions. This pause occurs during the transfer of the analog detector data from the scan unit to the host PC and also during the data type conversion, both of which occur in millisecond time scales. These burn spots can be mitigated by converting the data type after all frames are acquired and the information is saved, at the expense of increasing the amount of data on the host PC. An alternative option is to drive a beam blanker (with a shorter time response than the dwell time) by frame synchronization to blank the beam during the time the probe is paused before the start of the next frame.”

Further, I would find it very informative to have a better comparison with other existing charge mitigation techniques like variable pressure or gas injection systems. I understand that this is probably not achievable in their system, but a more extensive discussion on this point would be helpful.

It would indeed be interesting to compare the other methods to mitigate charging during SEM imaging such as gas injection-based charge compensation and variable pressure SEM which are mentioned in the introduction (lines 47-49 of the original manuscript, references 9-11). As the focus of this work is on cryogenic samples, we would like to highlight that there is no evidence that these approaches could be easily applied without interfering negatively on the cryopreservation or contamination rate.

3. Data Analysis:

The statistical significance of the improvements in image quality could be better quantified and I would attempt to see if metrics for determining optimal imaging conditions for different sample types could be available. For example the authors may use a FRC to quantify the image stability and improvement in relation to different amount of charging. similarly to what is commonly done in super-resolution microscopy.

The images obtained using the different scanning and integration strategies present a range of charging artefacts which could lead to completely dark images (*i.e.* raster frame integration). We have attempted to use single image FRC to compare the image quality between the different acquisitions; however, because single image FRC is a cross-correlation based approach, our results indicate that images with charging artefacts have at least as good resolution as the images with clearly reduced charging artefacts (Figure 2 below). Additionally, charging artefacts could lead to different types of histograms, skewed, multimodal, etc. As an example, in Figure 3 below we show three different acquisitions and their corresponding histograms. For interleaved scanning the histogram fits a

gaussian distribution, while for raster frame integration it is a skewed gaussian while for line integration the histogram is a bi modal distribution. We are not aware of a specific metric in the literature that is robust across the different distributions to compare the image quality. Therefore, in the manuscript, we present two metrics. One simply based on a mean histogram analysis (Figure 2, Supplementary Figure 4, 7, Supplementary Tables 1 and 2) since large dark regions would induce a shift of the histograms closer to the zero grey level. The second metric is related to the ability to pick undeformed biological features (Supplementary Figure 8).

Response Figure 3: Single image Fourier Ring Correlation (FRC) analysis. Mouse brain tissue imaged using 1.2 kV, 6.25 pA, 100 ns x100 repeats. The colour scheme relates to the locally measured resolution across the field of views using the FRC approach presented as overlays of the sample and the generated heat map (colour code on the left). It is observed that the overall FRC between the different imaging conditions are not statistically different (0.5%) despite visible charging artefacts.

Response Figure 4: histogram skewness analysis. Top row: Mouse brain tissue imaged using 1.2 kV, 6.25 pA, 100 ns x100 repeats. Middle row: Histogram skewness is quantified by the deviation between the histogram mean and the median, known as the Pearson's second skewness coefficient (absolute values are given here). A value of 0.11 skewness corresponds to the normal distribution (left), while a value of 0.89 to the skewed distribution (middle). Although a skewness can be calculated for the bimodal distribution (right), non-obvious skewed shape is observed, and the distribution shape must be considered. Bottom row: Therefore, we simply used the mean value to qualitatively indicate the presence of charging artefact (Supplementary Figure 7 and the overlay panel in this figure).

4. Technical Implementation: The integration of the external scan engine with existing microscope systems could be explained in more detail, The authors may provide better discussion in regards to compatibility issues with different microscope manufacturers and a clear guidance on how to implement this technique in laboratories with different equipment setups.

Most of the requests I made can be handled with text improvement and are only aimed at improving the overall output, The only two real experiments/analyses I am asking for are: the FRC or any other quantification of the image quality and stability and testing different scanning patterns, whose benefit can be in principle directly quantified with the FRC.

Happy to discuss this in further detail if required.

Alex de Marco

This is similar to a comment made by Reviewer #1. We have modified the Materials and Methods section (see earlier in this response letter) and added a supplementary figure (now Supplementary Figure 11) to provide more details on the installation of the scan engine on our system. Overall, it took approximately 96 hours to have the system up and running for single image acquisitions and an additional 48 hours for integration in automated image acquisitions (*ie* SerialFIB or automated slice and view).

In response to the comments raised we have also added new citations as below.

16. Liberman, L., Kleinerman, O., Davidovich, I. & Talmon, Y. Micrograph contrast in low-voltage SEM and cryo-SEM. *Ultramicroscopy* **218**, 113085 (2020).
72. Arat, K. T. *et al.* Charge-induced pattern displacement in E-beam lithography. *Journal of Vacuum Science & Technology B, Nanotechnology and Microelectronics: Materials, Processing, Measurement, and Phenomena* **37**, (2019).
73. Arat, K. T., Klimpel, T. & Hagen, C. W. Model improvements to simulate charging in scanning electron microscope. *Journal of Micro/Nanolithography, MEMS, and MOEMS* **18**, 1 (2019).
74. Jannis, D., Velazco, A., Béch e, A. & Verbeeck, J. Reducing electron beam damage through alternative STEM scanning strategies, Part II: Attempt towards an empirical model describing the damage process. *Ultramicroscopy* **240**, 113568 (2022).
75. Ichinokawa, T., Iiyama, M., Onoguchi, A. & Kobayashi, T. Charging Effect of Specimen in Scanning Electron Microscopy. *Jpn J Appl Phys* **13**, 1272–1277 (1974).

In addition to the above modifications, we have rephrased some of the sentences and addressed other typos in the main and supplementary documents. These corrections do not modify the content.

Reviewer #1 (Remarks to the Author):

The authors modified their manuscript according to my suggestions. I am pleased with the manuscript in its present form, and recommend it is published in Nature Communications.

Reviewer #2 (Remarks to the Author):

The manuscript has greatly improved.

Still I cannot SEE MCS nor mitochondria on the images of the manuscript I have at hand. Mitochondria are expected and most likely but cannot clearly be addressed at this resolution. The MCS cannot be resolved at the resolution used in this research, in fact they can only be identified at resolutions achieved by cryoTEMs.

In MCS, the membranes of two different organelles come into close proximity, typically within 10-35 nm of each other, without fusing. As the pixel size we have used here is 6.34nm we could consider that we are within range. However, we understand that confirmation of MSC should also come with the detection of specific protein complexes that we do not observe. Consequently, we have tone down all reference to MCS.

Reviewer #3 (Remarks to the Author):

I am happy with the revision,
nice work.

Alex